

# Estimating body volumes and surface areas of animals from cross-sections

Ruizhe Jackevan Zhao

Department of Mathematics, Northwest University, Xi'an, China

## ABSTRACT

**Background:** Body mass and surface area are among the most important biological properties, but such information is lacking for some extant organisms and most extinct species. Numerous methods have been developed for body size estimation of animals for this reason. There are two main categories of mass-estimating approaches: extant-scaling approaches and volumetric-density approaches. Extant-scaling approaches determine the relationships between linear skeletal measurements and body mass using regression equations. Volumetric-density approaches, on the other hand, are all based on models. The models are of various types, including physical models, 2D images, and 3D virtual reconstructions. Once the models are constructed, their volumes are acquired using Archimedes' Principle, math formulae, or 3D software. Then densities are assigned to convert volumes to masses. The acquisition of surface area is similar to volume estimation by changing math formulae or software commands. This article presents a new 2D volumetric-density approach called the cross-sectional method (CSM).
**Methods:** The CSM integrates biological cross-sections to estimate volume and surface area accurately. It requires a side view or dorsal/ventral view image, a series of cross-sectional silhouettes and some measurements to perform the calculation. To evaluate the performance of the CSM, two other 2D volumetric-density approaches (Graphic Double Integration (GDI) and Paleomass) are compared with it.
**Results:** The CSM produces very accurate results, with average error rates around 0.20% in volume and 1.21% in area respectively. It has higher accuracy than GDI or Paleomass in estimating the volumes and areas of irregular-shaped biological structures.
**Discussion:** Most previous 2D volumetric-density approaches assume an elliptical or superelliptical approximation of animal cross-sections. Such an approximation does not always have good performance. The CSM processes the true profiles directly rather than approximating and can deal with any shape. It can process objects that have gradually changing cross-sections. This study also suggests that more attention should be paid to the careful acquisition of cross-sections of animals in 2D volumetric-density approaches, otherwise serious errors may be introduced during the estimations. Combined with 2D modeling techniques, the CSM can be considered as an alternative to 3D modeling under certain conditions. It can reduce the complexity of making reconstructions while ensuring the reliability of the results.

Corresponding author
Ruizhe Jackevan Zhao,
JackevanChaos@outlook.com

# INTRODUCTION

Body mass and surface area are associated with many biological properties, including physiology, ecology, and evolution (*Sato et al., 2006*; *McClain & Boyer, 2009*; *Benson et al., 2017*; *Kinoshita et al., 2021*). Accurate estimates of these two values are often needed because unreliable results can lead to serious errors in subsequent research (*e.g.*, metabolic rate and speed calculations, *Motani, 2002*; *Sato et al., 2009*). However, body masses are unavailable for many large extant animals and most extinct organisms. Surface area information is also lacking because area can not be measured directly. Previous researchers have developed numerous approaches to solve this problem.

In general, there are two categories of approaches for body mass estimation: extant-scaling approaches and volumetric-density approaches (*Campione & Evans, 2020*). Extant-scaling approaches utilize skeletal measurements to reveal their relationships with body mass using regression equations (*Campbell & Marcus, 1992*; *Campione et al., 2014*). A classic and widely used example of extant-scaling approaches is the equation for quadruped mass based on humeral and femoral circumferences (*Anderson, Hall-Martin & Russell, 1985*).

The workflow of volumetric-density approaches is to first create a reconstruction of the animal being studied, then its volume is obtained, and an overall density is assigned to covert volume to mass (*Hurlburt, 1999*; *Henderson, 1999*; *Motani, 2001*). They have a much longer history than extant-scaling approaches, and numerous types of reconstructions have been developed over the past century, from physical models to 2D images to 3D virtual models. The earliest volumetric-density approaches were based on physical models. *Gregory (1905)* immersed a *Brontosaurus* model in water and acquired its volume using Archimedes' Principle, then he scaled the result to obtain the final estimate.

Some mathematical methods were developed later to calculate volume and surface area from 2D images. The first 2D volumetric-density method, Graphic Double Integration (GDI), was invented and introduced by *Jerison (1969)*. *Hurlburt (1999)* reviewed this method and presented detailed principles and computational steps. Protruding structures such as limbs or horns are first separated from the animal's main body, after which the latter is sliced equally into several parts. Each part is treated as a cylinder with elliptical bases. The semi-major and semi-minor axes of the two bases of each part are measured, then the average values are taken. By using the corresponding formulae or approximation formulae, the volume or lateral area of each elliptical cylinder can be calculated. Then the total volume or area is determined by summing the values of all component cylinders. Although GDI is not mathematically rigorous, it proves to have high accuracy (>95%) when dealing with objects with near elliptical cross-sections (*Jerison, 1969*).

*Henderson (1999)* developed a more rigorous math method called mathematical slicing to calculate volume and center of mass. This method also assumes that biological

cross-sections can be approximated by ellipses. To enable surface area calculation, *Henderson (2013)* extended mathematical slicing by decomposing these ellipses into multiple sets of points that divide the surface of the animal into numerous quadrilaterals. The area of each quadrilateral can then be calculated using vector cross products.

*Motani (2001)* noticed that some biological cross-sections in nature can not be well represented by ellipses. Due to the presence of such objects, Motani developed Paleomass, a program that brackets the true shapes using superellipses. The formula describing a superellipse is

$$\left|\frac{x}{a}\right|^k + \left|\frac{y}{b}\right|^k = 1 \qquad (1)$$

where *a* and *b* are semi-major and semi-minor axes respectively. It is noteworthy that this formula represents an ellipse when *k* equals 2. It was recently implemented in R and represents the latest study on 2D volumetric-density approaches (*Motani, 2023*). The new version can read bitmaps and generate a superellipse for every pixel along the sagittal axis. All superellipses are then combined into a 3D mesh, then Paleomass can calculate the cubic pixels (for volume) or square pixels (for surface area) that the mesh contains (*Motani, 2023*). The strength of this method is that it can generate intervals to bracket true animals with different k-values. Paleomass is good at estimating the volumes and surface areas of marine vertebrates, and it uses an equation for hydrodynamic foils, which includes a parameter that controls relative thickness, to approximate their fins (*Motani, 2023*).

With the rise of computer technology, three-dimensional modeling has been widely applied in animal reconstructions (*Bates et al., 2009*; *Eriksson et al., 2022*; *Segre et al., 2023*). The first step in 3D reconstruction of extinct vertebrates is to obtain the skeleton, which can be converted from photographs or 3D scans, then soft tissue is added to the skeleton. Once the reconstruction is accomplished, the volume and surface area of the 3D model can be acquired instantly using software. In addition, if a density distribution is assigned, 3D software can be applied to determine the location of the center of mass. A recent example is the high-resolution model of *Spinosaurus aegyptiacus* created by *Sereno et al. (2022)*. During the process of soft tissue reconstruction, errors and subjectivity can not be avoided (*Campione & Evans, 2020*). *Sellers et al. (2012)* invented the minimum convex hulling method, which generates minimum convex hulls to envelope the skeleton and adjusts the amount of soft tissue based on extant mammals. This method can reduce the errors introduced by soft tissue reconstructions, but it has the disadvantage of requiring numerous extant organisms as samples (*Motani, 2023*). Compared with 2D approaches, three-dimensional modeling requires proficient use of 3D software and is more time-consuming, so there is still a need to develop 2D methods.

This article presents a new 2D volumetric-density approach called the cross-sectional method (CSM). The CSM is a flexible approach that can handle any shape and can be applied to both extant and extinct animals. It processes gradually changing cross-sections directly and produces estimates with high accuracy. Combined with 2D modeling techniques, this method can be regarded as an alternative to 3D modeling in some cases. It can reduce the complexity of constructing animal models while ensuring the reliability of

the results. Elliptical or superelliptical approximations of biological cross-sections, which are assumed in some other 2D volumetric-density approaches, are shown here to possess limited validity under certain conditions.

## MATERIALS AND METHODS

Portions of this text were previously published as part of a preprint (https://www.biorxiv.org/content/10.1101/2023.10.13.562315v1).

### Data collection

To enable the estimation of body volume and surface area, some data are taken from the animal model being studied. The CSM requires one side view or dorsal/ventral view image and a series of cross-sectional profiles to perform computation. Figure 1 shows the workflow to collect data from the 3D model of a humpback whale (*Megaptera novaeangliae*) used in the validation tests of this study. In this particular case, the cross-sections required by the CSM are extracted from the 3D model, but in practice the cross-sections can be obtained in other ways (see the Working Examples section below). Protruding structures such as flukes, limbs and horns are first separated from the main body (Fig. 1B). Their volumes and surface areas can be calculated independently using the same method applied in the main body part. Then the side view (or dorsal/ventral view) outline of the animal under study is collected by drawing along the contour from photos, life reconstructions or orthogonal projections of 3D models (Fig. 1C).

The terms "slab" and "subslab" used by *Henderson (1999)* are inherited here. After the outline is obtained, the animal's profile is divided into several slabs using parallel lines (Fig. 1D). The portions of parallel lines truncated by the profile (*i.e.*, maximum heights in side views or maximum widths in dorsal/ventral views) are defined here as "identity segments".

After partitioning, each slab (except the first and last one) can be regarded as a frustum with parallel bases, which are probably different in shape. The slabs at two ends of the animal's sagittal axis can be regarded as cones with irregular-shaped bases. In the humpback whale example shown in Fig. 1, the tail fin is separated from the main body, hence only the anteriormost slab can be regarded as a cone. The next step is to collect the profiles of bases in each slab, which are the original body cross-sections of the animal (Fig. 1E). Then the area and circumference of each cross-section are acquired using image processing software.

### Body volume calculation

Consider a slab having two parallel bases which are different in shape (Fig. 2A). Each base has an identity segment (denoted by $d_0$ and $d_n$ respectively), which is used as a proxy for its area (denoted by $S_0$ and $S_n$ respectively). The ratio of $S$ to $d^2$ is defined and denoted by $\varphi$, *i.e.*,

$$\varphi = \frac{S}{d^2} \tag{2}$$
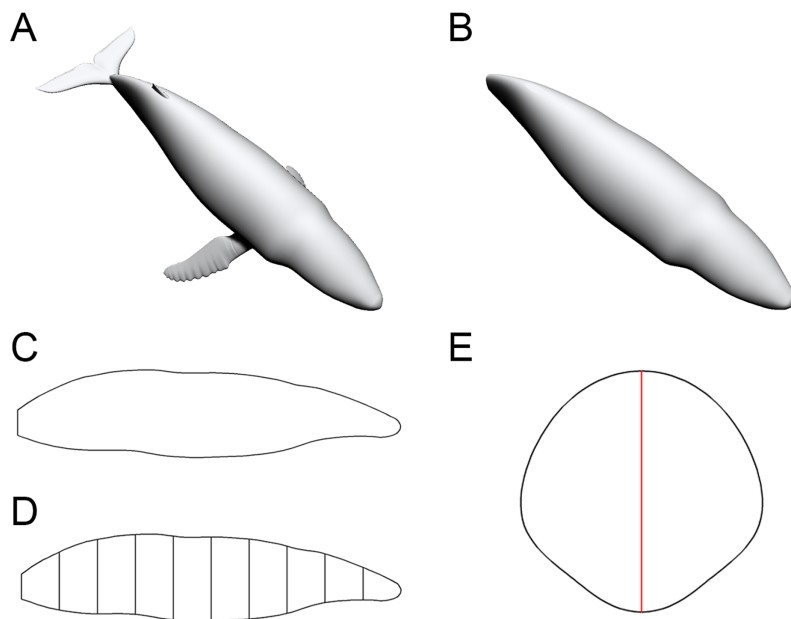

**Figure 1 Data collection process of the CSM.** (A) The 3D model of a humpback whale (*Megaptera novaeangliae*), from *Gutarra et al. (2022)*, published under the CC BY 4.0 license (https://creativecommons.org/licenses/by/4.0/). (B) Main body of the same model, with fins separated and removed. (C) Side view of the main body. (D) Side view of the main body after being sliced into 10 slabs. (E) One of the cross-sections of the main body, with the identity segment marked in red.

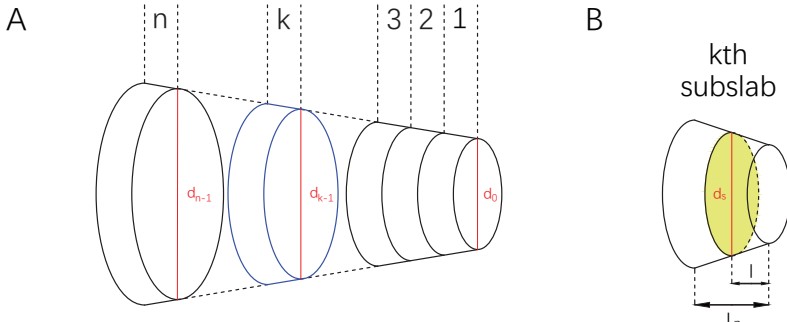

**Figure 2 Illustrations of a slab and subslabs.** (A) A slab equally partitioned into $n$ subslabs, with the $k$th subslab marked in blue and identity segments marked in red. (B) The $k$th subslab, with an arbitrary cross-section $B_s$ marked in green and its identity segment $d_s$.

Then slice the slab equally into $n$ subslabs with all the bases parallel to each other ($n$ is a positive integer). Now consider an arbitrary subslab, say the $k$th one (Fig. 2B). The upper base and lower base of the $k$th subslab are indexed by $B_{k-1}$ and $B_k$. The parameters (as defined above) of the lower base of the $k$th subslab are $d_k$, $S_k$, and $\varphi_k$ respectively. Total

length of the slab is denoted by $L$, and the length of each subslab is $L_n$. Assume that $\varphi_k$ follows a linear relationship from $\varphi_0$ to $\varphi_n$, then

$$\varphi_k = k\left(\frac{\varphi_n - \varphi_0}{n}\right) + \varphi_0. \tag{3}$$

Now consider the volume of the $k$th subslab. Length of identity segment $d$ can not be simply assumed to increase or decrease linearly, because maximum body heights/widths along an animal's sagittal axis often show irregular fluctuation. However, linearity is often used to approximate non-linearity at very small scales in calculus. If the partition of the slab is dense enough, it can be assumed that within each subslab $d$ also follows a linear relationship. Then for any cross-section (denoted by $B_s$) in the $k$th subslab parallel to the bases $B_{k-1}$ and $B_k$, it holds that

$$\varphi_s = \left(\frac{\varphi_k - \varphi_{k-1}}{L_n}\right)l + \varphi_{k-1} \tag{4}$$

$$d_s = \left(\frac{d_k - d_{k-1}}{L_n}\right)l + d_{k-1} \tag{5}$$

where $l$ is the distance from $B_s$ to $B_{k-1}$. Then let

$$\alpha_k = \frac{\varphi_k - \varphi_{k-1}}{L_n} \qquad \beta_k = \frac{d_k - d_{k-1}}{L_n}. \tag{6}$$

The area of cross-section $B_s$ can be calculated by

$$\begin{aligned}
S_s &= \varphi_s d_s^2 \\
&= (\alpha_k l + \varphi_{k-1})(\beta_k l + d_{k-1})^2 \\
&= \alpha_k \beta_k^2 l^3 + (2\alpha_k \beta_k d_{k-1} + \varphi_{k-1}\beta_k^2)l^2 \\
&\quad + (\alpha_k d_{k-1}^2 + 2\beta_k d_{k-1}\varphi_{k-1})l + \varphi_{k-1} d_{k-1}^2.
\end{aligned} \tag{7}$$

Then the volume of the $k$th subslab is

$$\begin{aligned}
V_k &= \int_0^{L_n} S_s \, dl \\
&= \frac{1}{4}\alpha_k \beta_k^2 L_n^4 + \frac{1}{3}(2\alpha_k \beta_k d_{k-1} + \varphi_{k-1}\beta_k^2)L_n^3 \\
&\quad + \frac{1}{2}(\alpha_k d_{k-1}^2 + 2\beta_k d_{k-1}\varphi_{k-1})L_n^2 + \varphi_{k-1} d_{k-1}^2 L_n.
\end{aligned} \tag{8}$$

In particular, if $\varphi$ is a constant (denoted by $\Phi$), then $\alpha_k = 0$ and

$$V_k = \frac{1}{3}\Phi\beta_k^2 L_n^3 + \beta_k d_{k-1}\Phi L_n^2 + \Phi d_{k-1}^2 L_n \tag{9}$$

The total volume of the slab is

$$V = \sum_{k=1}^{n} V_k. \tag{10}$$
The two slabs at both ends of the animal's sagittal axis are processed as slabs with constant $\Phi$ if they can be treated as cones, others are regarded to possess gradually changing cross-sections. The total main body volume can be acquired by summing the volumes of all the slabs. The volumes of structures separated (*e.g.*, fins, limbs) from the main body are calculated using the same method.

Some studies require the determination of the center of mass (CM) of an animal (*e.g.*, to find the balance point of animals, *Sereno et al., 2022*). To determine the vertical plane where the centroid of the $k$th subslab (denoted by $\overline{l_k}$) is located, the following formula can be applied:

$$\overline{l_k} = \frac{\int_0^{L_n} S_s l \mathrm{d}l}{\int_0^{L_n} S_s \mathrm{d}l}.$$  (11)

Then the plane containing the centroid of each slab (denoted by $\bar{x}$) can be determined by the following formula if a density distribution is developed

$$\bar{x} = \frac{\sum_{k=1}^n m_k x_k}{\sum_{k=1}^n m_k}$$  (12)

where $m_k$ is the mass of the $k$th subslab, and $x_k$ is the distance from the centroid of the $k$th subslab to the anteriormost base of the slab, *i.e.*,

$$x_k = \overline{l_k} + (k-1)L_n.$$  (13)

Equation (12) can also be extended to determine the location of the vertical plane containing the CM of the whole animal.

## Body surface area calculation

Similar method is applied to calculate the surface area. All parameters defined in volume calculation except $\varphi$ are inherited here. The circumferences of the upper base and lower base of the slab are denoted by $C_0$ and $C_n$. The ratio of $C$ to $d$ is denoted by $\psi$, *i.e.*,

$$\psi = \frac{C}{d}.$$  (14)

The parameters (as defined above) of the lower base of the $k$th subslab are $d_k$, $C_k$, and $\psi_k$. Assume that $\psi_k$ follows a linear relationship from $\psi_0$ to $\psi_n$, then it holds that

$$\psi_k = k\left(\frac{\psi_n - \psi_0}{n}\right) + \psi_0.$$  (15)

After slicing the slab equally into $n$ subslabs, linearity is used to approximate non-linearity at very small scales:

$$\psi_s = \left(\frac{\psi_k - \psi_{k-1}}{L_n}\right)l + \psi_{k-1}$$  (16)

$$d_s = \left(\frac{d_k - d_{k-1}}{L_n}\right)l + d_{k-1}$$  (17)

where $l$ is the distance from $B_s$ to $B_{k-1}$. Then let

$$\gamma_k = \frac{\psi_k - \psi_{k-1}}{L_n} \qquad \beta_k = \frac{d_k - d_{k-1}}{L_n}. \tag{18}$$

The circumference of cross-section $B_s$ can be calculated by

$$\begin{aligned}
C_s &= \psi_s d_s \\
&= (\gamma_k l + \psi_{k-1})(\beta_k l + d_{k-1}) \\
&= \gamma_k \beta_k l^2 + (\gamma_k d_{k-1} + \beta_k \psi_{k-1})l + \psi_{k-1} d_{k-1}.
\end{aligned} \tag{19}$$

Then the lateral surface area of $k$th subslab is

$$\begin{aligned}
A_k &= \int_0^{L_n} C_s \mathrm{d}l \\
&= \frac{1}{3}\gamma_k \beta_k L_n^3 + \frac{1}{2}(\gamma_k d_{k-1} + \beta_k \psi_{k-1})L_n^2 + \psi_{k-1} d_{k-1} L_n.
\end{aligned} \tag{20}$$

In particular, if $\psi$ is a constant (denoted by $\Psi$), then $\gamma_k = 0$ and

$$A_k = \frac{1}{2}\beta_k \Psi L_n^2 + \Psi d_{k-1} L_n. \tag{21}$$

The total lateral surface area of the slab is

$$A = \sum_{k=1}^n A_k. \tag{22}$$

The two slabs at both ends of the animal's sagittal axis are processed as slabs with constant $\Psi$ if they can be treated as cones, others are regarded to possess gradually changing cross-sections. The surface area of the main body is calculated by summing the lateral areas of all slabs. The surface areas of structures separated (*e.g.*, fins, limbs) from the main body are calculated using the same method.

### Validation and comparison

Four tests were performed on 3D models to verify the accuracy of the CSM. In all four tests, the volumes and surface areas of the models were first obtained using 3D software, and then the calculated results based on 2D methods were compared with these observed values. Only models accurately reproduced from museum mounts, photographs or 3D scans were used for validation (see *Gutarra et al., 2019*, *2022*, and http://digitallife3d.org/). The models created by the DigitalLife team contain some cavities for mouths and gullets in their head regions, which may introduce additional errors that affect the evaluation of the CSM. Therefore, the heads of them were separated and not included in the tests. Each model was scaled to 1 m in total length prior to the tests. To further evaluate the performance of the CSM, GDI and Paleomass were included for comparison, which approximate biological cross-sections with ellipses and superellipses respectively (*Hurlburt, 1999*; *Motani, 2023*). To ensure that the three methods could be compared within the same framework, twelve 3D models of extant or extinct aquatic animals were used (Fig. 3). Before the tests, protruding structures such as limbs, flukes and fins were

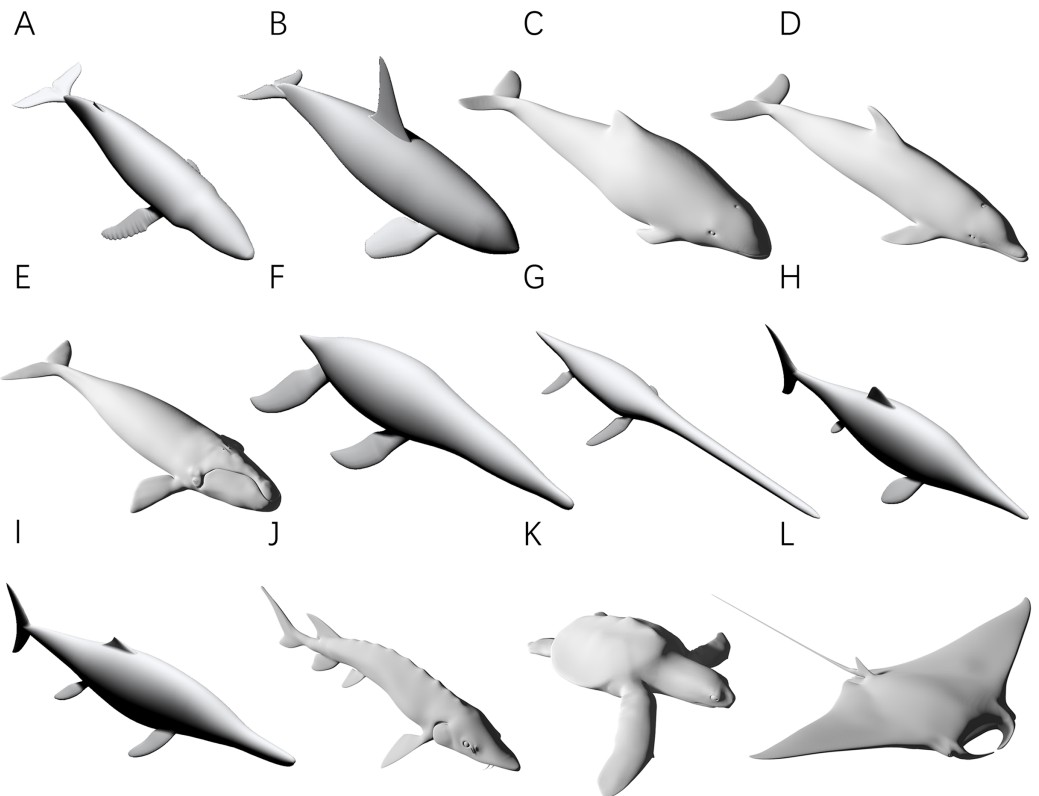

**Figure 3 3D models used for validation.** (A) Humpback whale (*Megaptera novaeangliae*). (B) Orca (*Orcinus orca*). (C) Harbor porpoise (*Phocoena phocoena*). (D) Bottlenose dolphin (*Tursiops truncatus*). (E) Southern right whale (*Eubalaena australis*). (F) *Liopleurodon*. (G) *Thalassomedon*. (H) *Ophthalmosaurus*. (I) *Temnodontosaurus*. (J) Atlantic sturgeon (*Acipenser oxyrhynchus oxyrhynchus*). (K) Hawksbill sea turtle (*Eretmochelys imbricata*). (L) Manta ray (*Mobula* cf. *birostris*). Image source: (A, B, F, G) are 3D models from *Gutarra et al. (2022)*, and (H and I) are from *Gutarra et al. (2019)*, all published under the CC BY 4.0 license (https://creativecommons.org/licenses/by/4.0). Other models are from https://sketchfab.com/DigitalLife3D, published under the CC BY-NC 4.0 license (https://creativecommons.org/licenses/by-nc/4.0/).

separated from the main body. Different structures from the same model may be used in different tests (see below).

The first test is to determine how many subslabs within a slab are required to obtain relatively accurate volume and surface area estimates. The calculation of the CSM uses linearity to approximate non-linearity at small scales, and the purpose of this test is to determine how dense the partition needs to be. The main bodies of the humpback whale (Fig. 3A), the orca (Fig. 3B), the *Liopleurodon* (Fig. 3F) and the Atlantic sturgeon (Fig. 3J) were selected for this test. Each model was treated as one slab and partitioned into 2–16 subslabs, with two increments each time. Two random cross-sections were taken from two ends of each model. After calculation, the results were compared with the values obtained using 3D software.

The purpose of the second test is to determine how many slabs are required to produce relatively accurate estimates. It is intuitive to expect that as the number of slabs increases, the error rate will decrease. The four models selected in the first test were also used in this

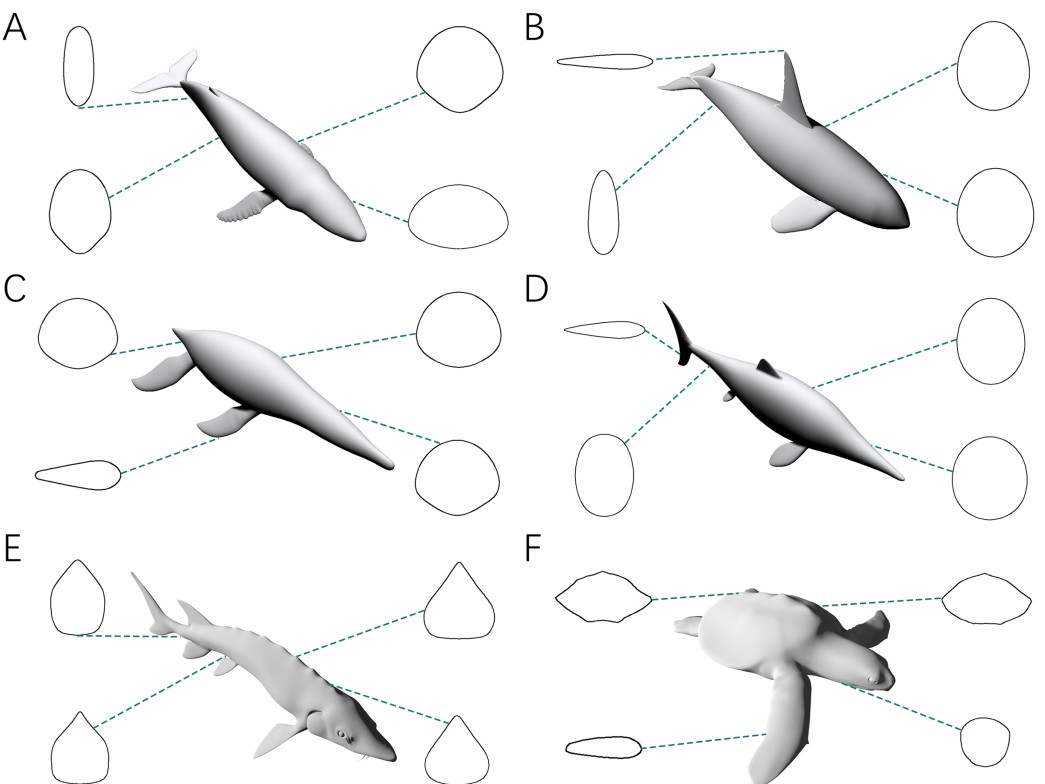

**Figure 4 Representative cross-sections of the animal models.** (A) Humpback whale (*Megaptera novaeangliae*). (B) Orca (*Orcinus orca*). (C) *Liopleurodon*. (D) *Ophthalmosaurus*. (E) Atlantic sturgeon (*Acipenser oxyrhynchus oxyrhynchus*). (F) Hawksbill sea turtle (*Eretmochelys imbricata*). Sizes of cross-sections not to scale. 3D models (A–C) are from *Gutarra et al. (2022)*, and (D) is from *Gutarra et al. (2019)*. (E and F) are from https://sketchfab.com/DigitalLife3D.

test. Each model was divided into 2–16 slabs, with two increments each time. Each slab was further sliced into 10 subslabs prior to computation.

The third test aims to find out whether the CSM has comparable or better performance than GDI or Paleomass in processing animals with near circular cross-sections. The main bodies of five cetaceans, two plesiosaurs and two ichthyosaurs (Figs. 3A–3I) were used in this test. The main body of each model has rounded or oval cross-sections, which can be well approximated by ellipses or superellipses (Figs. 4A–4D).

The fourth test is designed to demonstrate that the CSM can still accurately estimate the volumes and surface areas when dealing with irregular-shaped biological structures. The models used in this test include fins and flippers of secondarily aquatic tetrapods (Figs. 3B, 3F and 3H), the main body of the Atlantic sturgeon (Fig. 3J), the main body of the hawksbill turtle (Fig. 3K) and the pectoral fin of the manta ray (Fig. 3L).

Both the third and the fourth tests compare the performance of the CSM, GDI, and Paleomass. The criteria used in these two tests are described below. In GDI, each object was first equally sliced into 120 slabs, then the volume was calculated using the formulae proposed by *Hurlburt (1999)* after the necessary measurements were made. Paleomass was performed using the corresponding package in R (*Motani, 2023*). The four fin specimens

in the fourth test were treated as foils and the others were treated as main bodies (for detailed methods, see *Motani, 2023*). The k-value range was set to 2–2.3 in the third test. This is the suitable range for modern cetaceans (*Motani, 2023*), and it is assumed that the k-values of plesiosaurs and ichthyosaurs are also in this range. In the fourth test, the k-value range was set to 1.6–2.4, which successfully bracketed all the aquatic species tested by *Motani (2023)*. To enable the calculation of error rates and comparison with other methods, the average value of the upper and lower bounds provided by Paleomass was calculated for each model, following *Motani (2023)*. In the CSM, each object was equally sliced into 12 slabs and each slab was further divided into 10 subslabs, then the volume and surface area were calculated after parameters of the bases in each subslab were obtained.

After calculation, the error rates generated by different methods were compared. Error rate is defined as

$$\text{Error Rate} = \frac{\text{Calculated Value} - \text{Observed Value}}{\text{Observed Value}} \tag{23}$$

when the calculation underestimates the true value, the error rate is negative; when overestimating, the error rate is positive. The mean error is calculated as:

$$\text{Mean Error} = \frac{\sum |\text{Error Rate}|}{\text{Sample Number}}. \tag{24}$$

### Working examples

The validation procedure is based on 3D models, but in practice the cross-sections required by the CSM can be obtained in other ways. This section presents two working examples of the CSM to show how this method can be used to estimate the sizes of extant or extinct animals.

Example 1: bottlenose dolphin (*Tursiops truncatus*)

For extant animals, the cross-sections can be obtained by sawing dissection or CT scanning. Once the cross-sectional profiles of an animal have been successfully collected, the CSM can be used to estimate the masses and surface areas of conspecific individuals. A series of cross-sections of a female bottlenose dolphin are provided by *Huggenberger, Oelschläger & Cozzi (2018)*. The mass of the female is not recorded, so the cross-sections were applied to estimate the body mass of a male individual, of which the size information and side view photos are provided in the same publication. The female was sawn into 73 slices, and 12 of them were selected in this study as representative cross-sections. Correspondingly, the female was divided into 12 slabs, each containing six slices, except for the last one that contains seven slices. The slabs are not of equal length, possibly due to uneven sawing. The side view silhouette of the male was then sliced in the same way in 2D software (Fig. 5). The volume of the main body of the male was then calculated using the CSM. Side view photos or cross-sectional profiles of the fins are not provided by *Huggenberger, Oelschläger & Cozzi (2018)*, so relative sizes of the fins to the whole body were summarized from the 3D model used in the third test (Fig. 3D). To convert volume to mass, a mean density of 1,027 kg/m$^3$ was assigned following *Motani (2023)*.

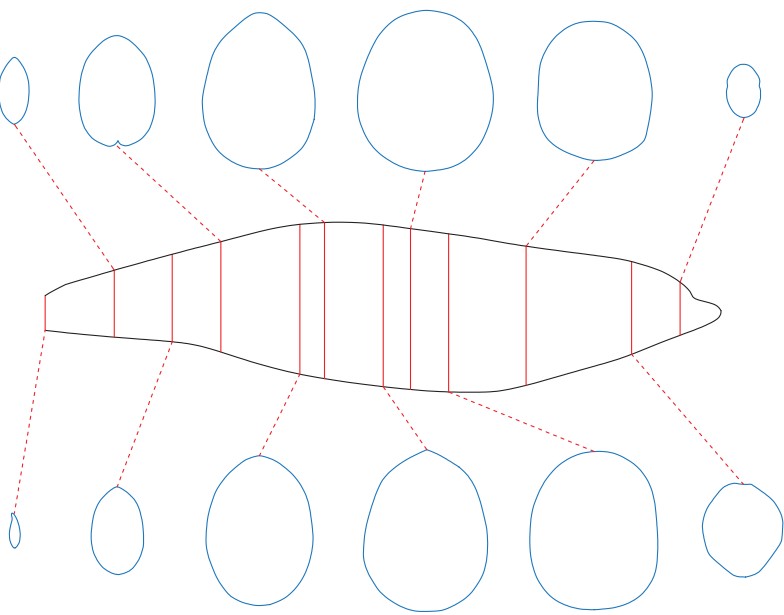

**Figure 5 The bottlenose dolphin selected as a working example.** Main body of the dolphin, with identity segments (red) and 12 representative cross-sections (blue).

Example 2: *Tyrannosaurus rex*

Together with 2D modeling techniques, the CSM can be applied to estimate the body size of extinct vertebrates. To demonstrate how this can be accomplished, I created a side view reconstruction of *Tyrannosaurus rex* AMNH 5027. Michael Joshua (M. Joshua, February 2024, personal communication) provided the 3D scan of this mount, which allows for precise measurements. Although this individual has already been mounted, the reliability of the mount can not be guaranteed. For example, the tail was restored to include 53 caudals (*Osborn, 1917*), which is unrealistic for *T. rex* (*Brochu, 2003*). Therefore, some changes should be made, and it is easier to accomplish this in a 2D environment than in 3D software. In addition, the CSM combined with 2D modeling can be applied to other extinct vertebrates that are not mounted. A side view reconstruction of AMNH 5027 and some cross-sections were created first (Fig. 6A; see the Supplemental Material for detailed process). There are active debates about how much soft tissue should be added when reconstructing extinct animals (*Bates et al., 2009*; *Hutchinson et al., 2011*; *Hurrell, 2019*; *Ibrahim et al., 2020*; *Sereno et al., 2022*). However, this issue is beyond the scope of this study, and barely any soft tissue was added to the ribcage region. *Persons & Currie (2011)* argued that the tail muscle *M. caudofemoralis* plays an important role in hindlimb retraction and has an impact on the position of the CM in theropods. Hence the tail muscles were reconstructed following their criteria. The cross-section of the hindlimbs was reconstructed following *Paul (1988)*. The soft contours of the forelimbs and toes of AMNH 5027 are unknown, and were approximated using cylinders. This would not significantly affect the final result due to their small sizes. The main body was separated into eight slabs (Fig. 6B), and each hindlimb was treated as a single slab. Each slab was divided into 1,000

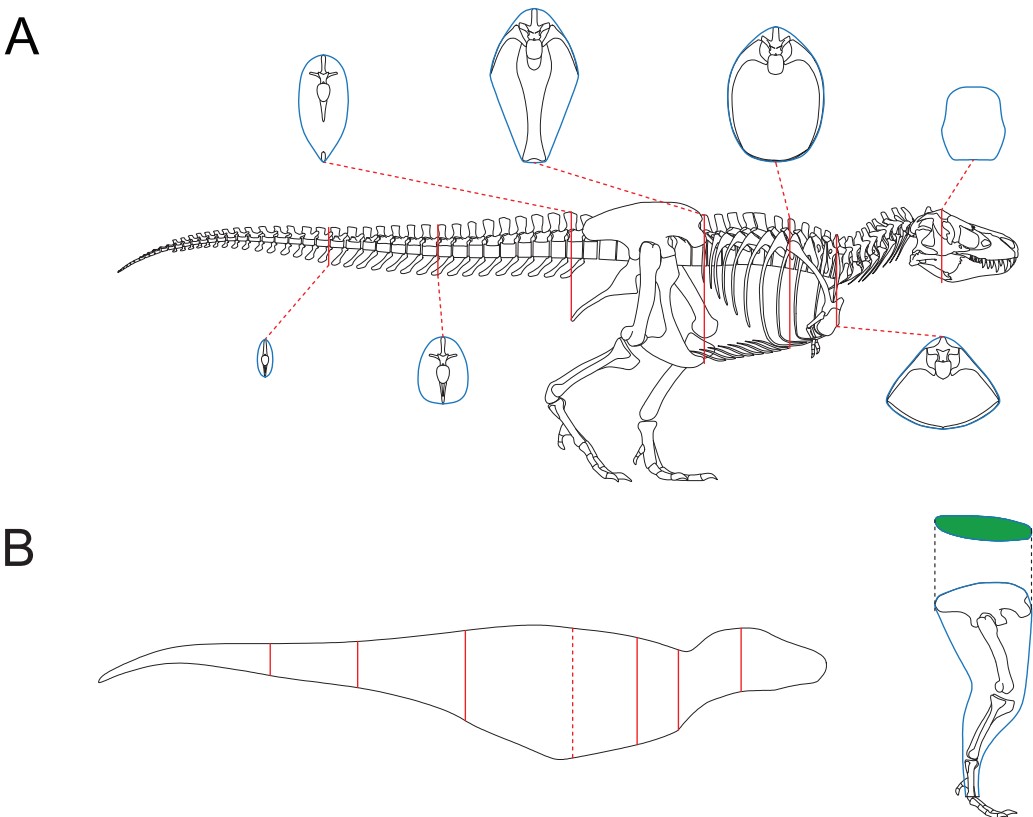

**Figure 6 Reconstruction of *Tyrannosaurus rex* AMNH 5027.** (A) Skeletal reconstruction and body cross-sections. (B) Main body of AMNH 5027 partitioned into eight slabs by the seven cross-sections (red). The anterior base of the fifth slab is represented by the red dashed line, which almost overlaps the vertical plane containing the CM. The posture of hindlimb was adjusted for soft tissue reconstruction. The cross-section of hindlimb (green) was reconstructed following *Paul (1988)*.

subslabs, and the CSM was used to calculate their volumes. It is possible that the result presented here underestimates the size of AMNH 5027, but the amount of soft tissue can be precisely adjusted in 2D software without much effort. In addition, some previous researchers have provided mass estimates for this individual based on skeletal reconstructions that also contain little soft tissue around the ribcage (*e.g.*, *Paul, 1988*). Therefore, the result presented in this article can be compared with those from previous studies. It is impractical for the CSM to determine the vertical planes containing the centroids of the limbs in this case, so only the CM of the main body was determined using Eqs. (11) and (12). *Henderson (1999)* argued that the CM of the main body would still be informative if the limbs are not included in the calculation. The size and position of the lungs were reconstructed according to the criteria proposed by *Henderson (1999)*, and the lungs were simplified into an ellipsoid to facilitate the determination of the centroid. The density of the lungs was set to 0, and that of the other body parts was set uniformly to 1,000 kg/m$^3$.

## Software application

All the 3D models used in the four validation tests were first processed in Rhino 7. Each model was divided using the *WireCut* command, then the volume and surface area of the selected part were acquired with the *Volume* and *Area* commands. Side view and dorsal/ventral view images of the separated models were obtained with *Make2D*. To generate the cross-sections required by the CSM, the *ClippingPlane* command was used.

Two-dimensional images were then imported into AutoCAD 2020, where they were sliced into slabs or subslabs. Measurements of each slab or subslab were taken and exported to Excel, where the calculations of GDI and the CSM were finally performed. To ensure that future users can replicate the CSM, a step-by-step tutorial can be found (*Zhao, 2024*). The photos of the bottlenose dolphin selected as a working example were loaded into AutoCAD, then the side view and cross-sectional silhouettes were created. The *T. rex* model was also constructed in AutoCAD.

Both Rhino and AutoCAD are high-precision industrial software. They have been used in previous studies to estimate the body sizes of extinct animals and have shown good performance (*e.g.*, *Henderson, 2006*; *McHenry, 2009*).

Paleomass implemented in R requires bitmaps (*Motani, 2023*), so the two-dimensional images were exported from AutoCAD as PNGs. Each PNG was set to possess 6,000 × 4,000 pixels as suggested by *Motani (2023)* for better performance. The images were imported into Photoshop 2020 for editing and then imported into R 4.1.3 (*R Core Team, 2022*) for the final calculations.

## RESULTS

Error rates of the four tests are presented in Figs. 7, 8, Tables 1, 2 respectively. The detailed results can be found in the Supplemental Material.

Figure 7 shows the results of the first test. It demonstrates that the error rates tend to stabilize for both volume and area estimations when the number of subslabs reaches a certain value: eight for the humpback whale, six for the orca, six for the *Liopleurodon* and 10 for the sturgeon. The second test similarly shows that the error rates tend to stabilize when the number of slabs reaches a certain value if each slab is equally divided into 10 subslabs: six for the humpback whale, 12 for the orca, 12 for the *Liopleurodon* and 12 for the sturgeon.

Table 1 shows the results of the third test. Models that are not bracketed by Paleomass are marked by an asterisk (*). All the three validated methods show good performance, with error rates below 5% on average. This corroborates the validity of 2D volumetric-density methods when dealing with animals with rounded or oval cross-sections, as shown in previous studies (*Henderson, 1999*; *Motani, 2023*). For both volume and surface area calculations, the CSM has similar or slightly higher accuracy than GDI and Paleomass.

In the fourth test, the error rates of GDI and Paleomass increase significantly (Table 2). This indicates that an elliptical approximation, as assumed in GDI, is not suitable for all biological cross-sections. Paleomass treats the four fin/flipper samples as foils, which are described by an equation with one variable that controls relative thickness (t-value, see

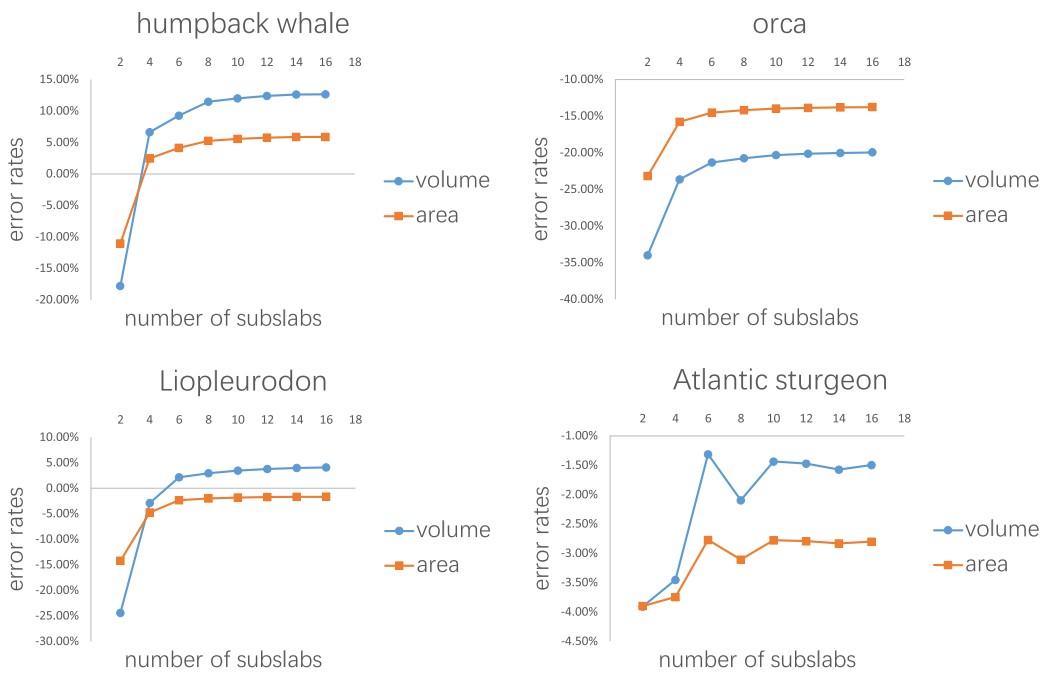

**Figure 7 Results of the first test.** Each model tested was treated as one slab and partitioned into 2–16 subslabs.                                           

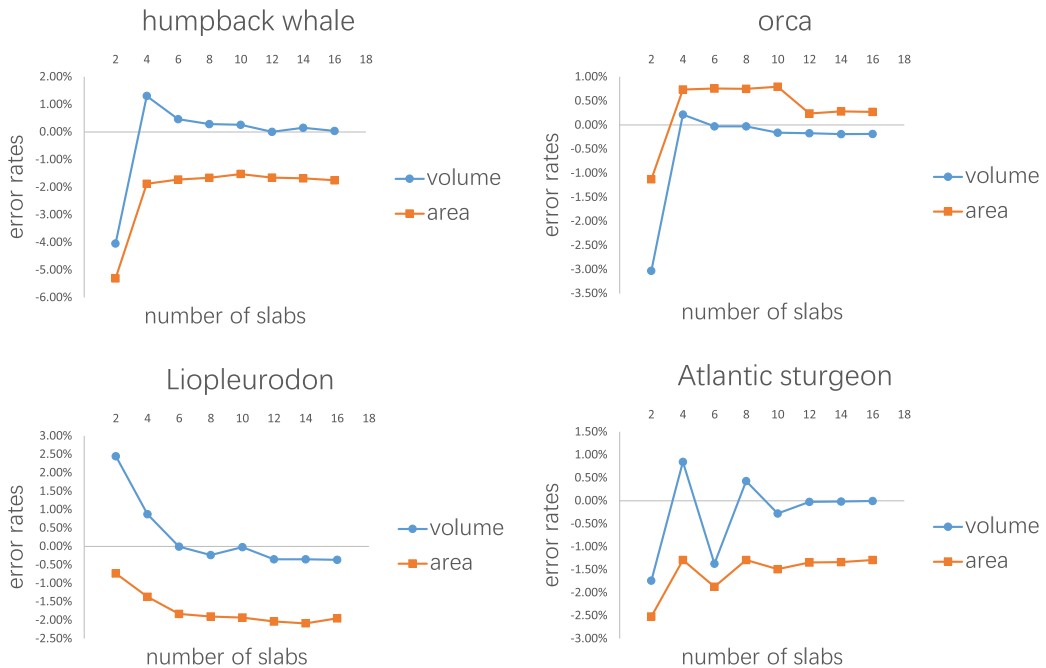

**Figure 8 Results of the second test.** Every model tested was sliced into 2–16 slabs, each of which was further divided into 10 subslabs.                                           
**Table 1 Results of the third test.** Models that aren't bracketed by Paleomass are marked with an asterisk (*).

| Model | Volume | | | Surface area | | |
|---|---|---|---|---|---|---|
| | GDI | Paleomass | CSM | GDI | Paleomass | CSM |
| Humpback whale | 4.40% | 6.77%* | 0.04% | 0.47% | 6.25%* | −1.66% |
| Orca | −0.20% | 2.05% | −0.17% | 1.34% | 3.56% | 0.24% |
| Harbor porpoise | 0.69% | 2.50% | −0.28% | −0.87% | 0.02% | −0.83% |
| Bottlenose dolphin | 2.45% | 3.31% | 0.03% | −0.03% | 1.01% | −0.63% |
| Southern right whale | 6.43% | 7.90%* | 0.10% | 1.75% | 2.92%* | −1.13% |
| *Liopleurodon* | 2.94% | 5.33%* | −0.35% | −0.82% | 2.46%* | −2.04% |
| *Thalassomedon* | −1.18% | 0.91% | −0.14% | −1.89% | 0.38% | −1.29% |
| *Ophthalmosaurus* | −1.20% | 0.64% | 0.89% | −1.68% | 0.31% | −1.28% |
| *Temnodontosaurus* | −0.34% | 2.02% | 0.09% | −1.44% | 1.04% | −1.37% |
| Mean | 2.20% | 3.49% | 0.23% | 1.14% | 1.99% | 1.15% |

**Table 2 Results of the fourth test.** Models that aren't bracketed by Paleomass are marked with an asterisk (*).

| Model | Volume | | | Surface area | | |
|---|---|---|---|---|---|---|
| | GDI | Paleomass | CSM | GDI | Paleomass | CSM |
| *Liopleurodon* flipper | 16.79% | 24.16% | −0.22% | 8.74% | 4.30% | 0.14% |
| Orca dorsal fin | 12.10% | −29.55% | −0.20% | 7.84% | −12.94% | −1.13% |
| *Ophthalmosaurus* tail fin | 14.18% | 15.44% | −0.40% | 7.72% | 0.74% | −1.36% |
| Manta ray pectoral fin | 16.77% | −57.32% | 0.04% | 6.62% | −12.38% | −1.86% |
| Atlantic sturgeon main body | 12.64% | 9.84%* | −0.03% | −0.54% | 5.88%* | −1.35% |
| Hawksbill turtle main body | 18.69% | 14.43%* | 0.08% | 6.16% | 8.06%* | −1.34% |
| Mean | 15.20% | 25.12% | 0.16% | 6.27% | 7.39% | 1.20% |

*Motani, 2023*). However, high error rates occur in the estimated results from Paleomass in these samples. Paleomass also fails to bracket the Atlantic sturgeon and hawksbill turtle with the selected k-value range (1.6–2.4). The CSM generally has much better performance than GDI or Paleomass in the fourth test, with error rates always lower than 2%.

The volume of the male bottlenose dolphin was estimated to be 0.3098 m$^3$. Assuming an overall density of 1,027 kg/m$^3$ following *Motani (2023)*, the body mass is around 318 kg. This suggests that the method used here overestimates the mass by 10.42%, which is much higher than the error rates in the validation tests. However, such a high error rate is probably caused by sexual dimorphism (see Discussion).

The calculated volume of the *T. rex* AMNH 5,027 is 7.7249 m$^3$. Applying the density distribution assigned above, the estimated body mass is 6,952 kg. This result is within the range of 5,709~7,908 kg provided by previous studies (*Paul, 1988*; *Henderson, 1999*; *Seebacher, 2001*; *Therrien & Henderson, 2007*), based on skeletal reconstructions of this
individual that also contain little soft tissue around the ribcage. The vertical plane containing the CM is located at the anterior edge of the pelvic girdle and almost overlaps the anterior base of the fifth slab (Fig. 6B). This result is consistent with the estimated CM of *T. rex* from previous studies (*e.g.*, *Henderson, 1999*; *Bates et al., 2009*; *Snively et al., 2019*).

## DISCUSSION

The rationale for including extinct animals in the tests merits a discussion. The body outlines of most extinct animals are unknown, and the 3D models remain interpretive reconstructions. But body volumes and surface areas are always obtained after the models are constructed in all volumetric-density methods. In other words, all volumetric-density methods actually estimate the volumes and surface areas of the models rather than the true animals. The inclusion of extinct animals in the tests successfully demonstrates that the CSM can provide accurate volume and area estimates of artificial models if enough cross-sections are included in the computation. Thus it is a flexible method that can be applied to extant or extinct animals.

Although the validation of the CSM in this article is based on 3D models, the acquisition of cross-sections in the actual application process can be accomplished by other means, which are shown in the Working Examples section. For extant animals, the cross-sections can be obtained through dissection or CT scanning. Although the bottlenose dolphin selected as an example can be weighed directly, this method can be extended to other animals, including giant whales that weigh tens of tons. If the dissection of one captured or naturally deceased individual sheds light on the cross-sectional profiles, the CSM can be used to study the body sizes of wild populations. In such cases, the CSM requires only one dorsal/ventral or side view photo rather than two needed in other 2D volumetric-density approaches, and the results are more accurate. Although the error rate was relatively high (10.42%) in the case of the bottlenose dolphin, this error may be partly caused by sexual dimorphism. Female bottlenose dolphins were found to be heavier than males of the same body length (*Mallette et al., 2015*). The male used as an example is 284 cm in body length, and the estimated body mass of a female at this length is 304.8 kg using the regression equation provided by *Mallette et al. (2015)*. This would reduce the error rate to 4.33%. Therefore, if there is a significant sexual dimorphism in the species being studied, it is recommended to obtain different sets of cross-sections for males and females respectively.

The *T. rex* example shows how the CSM can be applied to extinct animals. In the study of vertebrate paleontology, the CSM, coped with 2D modeling, can be regarded as an alternative to 3D modeling in some cases. If the user has sufficient knowledge of the skeletal anatomy of the animal under study, the cross-sections can be constructed accurately in 2D software. Although little soft tissue was added to the ribcage of the *T. rex* model constructed in this study, it is conceivable that controlling for the amount of soft tissue would be easier in a 2D environment than in 3D software.

Although the results of the first and second tests provide guidelines for the CSM in real-world applications to a certain extent, they are based on limited samples. The results of the first test show that 10 subslabs for each slab is adequate to satisfy the linear assumption, but future users can choose a number of subslabs larger than this order of
magnitude (*e.g.*, 1,000, as used in the *T. rex* example), which can be easily accomplished in CAD software like AutoCAD. Figure 8 shows that the number of slabs needed for the error rate to stabilize varies across species. Rather than suggesting a definitive protocol for how many cross-sections should be obtained for each animal, I recommend acquiring a cross-section wherever there is a significant change in shape. The performance of the CSM would be improved if the user have sufficient knowledge of the variation in cross-sectional shapes of the animal under study. In addition, the distances between the cross-sections do not have to be equal, as shown in the two working examples. The amount of cross-sections acquired can be appropriately reduced for regions where shape changes are not significant.

The purpose of the comparison of the three methods (GDI, Paleomass, and the CSM) is to demonstrate the limitations of elliptical or superelliptical approximation. It has long been assumed that the cross-sections of an animal's main body or limbs can be approximated by ellipses (*Campione & Evans, 2020*). For some species with rounded or oval cross-sections, elliptical approximation has good performance (Table 1).

*Motani (2001)* noticed that some cross-sections in nature can not be well represented by ellipses. This is supported by some of the samples tested in this study (Figs. 4E, 4F). Due to the presence of such objects, the application scope of elliptical approximation methods such as GDI is limited. Paleomass also produces significant errors in the fourth test. The strength of Paleomass is its ability to bracket the true shape using superellipses with different k-values (*Motani, 2023*). However, some subsequent research requires point estimates rather than intervals (*e.g.*, kinematic analysis, *Sato et al., 2006*). Taking the average value may be an option, but this would be identical to approximating the shape with a particular superellipse. Instead of superelliptical bracketing, Paleomass treats the fins of animals as hydrodynamic foils, but a single formula with only one variable controlling the thickness may not be sufficient to describe all types of fins and flippers.

The CSM presented in this article calculates volumes and surface areas from cross-sectional profiles directly rather than approximating. Instead of testing the performance of the CSM on complete models, irregular-shaped biological structures were separated and tested independently. This is because such structures (*e.g.*, fins in aquatic animals) are sometimes so small that errors in them have little impact on the overall accuracy. In the third and fourth tests, each model was sliced into 12 slabs and each slab was further divided into 10 subslabs, because they are the minimum values that stabilize the estimates for all the models used in the first and second tests. Under this criterion, the CSM produces more accurate estimates for volume and area than GDI or Paleomass in dealing with irregular-shaped structures. Processing profile images may introduce additional errors, but the total error rates are around or below 2% for all the samples tested. Unlike many previous 2D volumetric-density approaches which assume a constant superelliptical k-value (k = 2 for ellipse) along the sagittal axis, this method is more flexible by assuming and handling gradually changing cross-sections. It generates point estimates rather than intervals, so that the results can be directly incorporated in subsequent studies such as scaling regressions (see "hybrid approaches" in *Campione & Evans, 2020*).

Despite its accuracy, the CSM has some drawbacks and limitations. Compared to other 2D volumetric-density methods, it requires more preparation before calculation. The CSM

requires a series of cross-sectional profiles, which can be time-consuming to acquire. To simplify the calculation, the CSM does not introduce a coordinate system, which makes it impractical to calculate the detailed location of the CM of an animal (*i.e.*, only the vertical plane containing the CM can be determined). In addition, the calculation process requires the cross-sections obtained being parallel to each other. When dealing with curved objects, it is recommended to acquire the vertical cross-sections. This issue can be avoided if the animal under study has been subjected to a CT scan since in most cases the cross-sections obtained in this way are vertically oriented. For extinct vertebrates, previous researchers have proposed methods to determine the vertical cross-sections of their ribcages (*e.g.*, *Welles, 1943*; *Hirasawa, 2009*; *Richards, 2011*; *O'Keefe et al., 2011*). Alternatively, the users can straighten the animal model prior to calculation. A representative example is provided by *Motani & Pyenson (2024)*. These two methods can enable the CSM to deal with curved objects, but they may take extra effort to implement.

*Paul (1997*, *2022)* suggested that accurate skeletal profiles are essential for reconstructing extinct or extant vertebrates, but a rigorous reconstruction of the ribcage has often been ignored or not published in previous studies. Careful examination of cross-sections is also advocated by other researchers (*e.g.*, *Motani, 2001*). It is suggested that future researchers pay more attention to detailed and careful reconstruction or acquisition of cross-sectional profiles because simply assuming an elliptical or superelliptical cross-section can lead to serious errors, as shown in this article.

## CONCLUSION

The cross-sectional method (CSM) is a new 2D volumetric-density approach, which processes cross-sectional profiles directly rather than approximating. The CSM requires a side view or dorsal/ventral view image and a series of cross-sectional silhouettes to perform calculation. It integrates biological cross-sections into volumes and surface areas, and produces point estimates with a high accuracy. Combined with 2D modeling, this method can be regarded as an alternative to 3D modeling in some cases. It can reduce the complexity of modeling while producing reliable results. Rather than assuming elliptical or superelliptical cross-sections empirically, future scholars are suggested to carefully examine the profiles to acquire the true shapes.

## ACKNOWLEDGEMENTS

I thank Beneden Parotodus (pseudonym, used as requested) for assessing math formulae before publication. Andrew Orkney and Frank Fang are thanked for advice on improving the manuscript. Michael Joshua and Frederick Dakota provided materials and suggestions for the *T. rex* reconstruction. The reviewers Philip Novack-Gottshall and Donald Henderson and the academic editor offered constructive comments to improve this study. I also thank Duncan Irschick and the DigitalLife team for access to their 3D models.

### Funding

The author received no funding for this work.

### Competing Interests

The author declares that they have no competing interests.

### Author Contributions

- Ruizhe Jackevan Zhao conceived and designed the experiments, performed the experiments, analyzed the data, prepared figures and/or tables, authored or reviewed drafts of the article, and approved the final draft.

### Data Availability

The raw data are available in the Supplemental Files and a tutorial on CSM is available at Zenodo: Ruizhe Jackevan Zhao. (2024). Cross-sectional Method (v1.1). Zenodo. https://doi.org/10.5281/zenodo.10974317.

### Supplemental Information

Supplemental information for this article can be found online at http://dx.doi.org/10.7717/peerj.17479#supplemental-information.

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
