# Peer review of "Estimating body volumes and surface areas of animals from cross-sections"

_PeerJ, doi:10.7717/peerj.17479_

## Round 0.1 · original submission · Major Revisions

You provide a new way to estimate body volumes and surface areas from cross-sections and test its performance. This approach is of interest for the community but there are crucial points which need to be addressed before publication. I apologize for the delay in getting my decision to you, but I was hoping to receive an additional review which got an extension but unfortunately did not arrive. However, I feel that the input of the reviewers and my own should be sufficient to revise the manuscript. The main points to be addressed are:

Background on competing approaches: I agree with reviewer 1 that more information and comparisons on the competing approaches needs to be provided for the readers not so familiar with these approaches in the introduction and discussion. See also the recommendations of reviewer 2 concerning particularly prior surface area estimations.

Validity of findings: Your test seem to produce valid results and comparisons seems adequate at first glance (compare reviewer 2) but the reasons for their performance are not fully explored. As pointed out by reviewer 1, the test of the methods lack generality and have inconsistencies. Please repeat test 1 by using more specimens, and using entire specimens as suggested by reviewer 1. Comparisons between methods have inconsistencies as more slabs are used in CS than other methods making it difficult to assess if it is truly a better methods or because specimens were able to be measured with more precision (compare reviewer 1).

Scientific reproducibility and accessibility: The new cross-sectional method is described in sufficient details to enable replication pending considerable effort.
You present a new method, but the software does not appear to be publicly available and no clear guidelines or limitations for its use are provided (compare reviewer 2). This makes it hard to reproduce your approach and widely apply it for other taxa in a straightforward way (compare reviewer 2). Particularly critical is the complexity of generating initial 3D models in an expensive and advanced software not accessible to all (e.g., Rhino; compare reviewer 2). Reviewer 2 also points out that Rhino already generate volumes and surface areas, which makes your approach a bit redundant unless you can make the entire process available in a publicly available software (e.g., Motani 2023 - which method you analyzed - makes his software available for use). As pointed out by reviewer 2 your approach could be easily extended by computing the centroid of the used 3D shapes or entire body region to locate the balance points or center of pressure for life calculations. The limitation of your approach that it depends on the body or body part being straight and how (if at all) it can be extended to cope with curved necks should be highlighted and discussed (compare reviewer 2).

Formatting and language issues: once revised, please check your manuscript for any inconsistencies (see suggestions by reviewers 1 and 2).

Figures: Please follow the recommendations of reviewer 2 to rework the dashed diagonals representing the diameters on Figure 2 and show the representative frustum and its subsection with same orientation as the body forms shown in figures 1 and 3. Please also follow the recommendation of reviewer 2 to present several sets of cross-sections from the full axial bodies and limbs from a limited selection of the models shown in figure 3 and to include in samples one each of a toothed and baleen cetacean, an ichthyosaur, a plesiosaur, the sturgeon and the turtle.

Please address these points as well as all other points raised including those in annotated pdfs.

I look forward to receiving the revised manuscript.

·

Basic reporting

Generally well written but would benefit from editorial assistance with a fluent English writer to clean up numerous typos, grammatical errors, and occasionally non-standard sentences. Background information on the two competing methods (GDI and Paleomass) is insufficient at introducing them to readers, with critical information placed instead in the Conclusions.

Experimental design

The overall design is fine. However, see below for concerns on the validity of the findings, which bear on the design.

Validity of the findings

Test 1 (how many slabs to cut the digital specimen into?) is lacking in generality; it should be repeated using more specimens, and using entire specimens. Tests 2 and 3 (how does CS compare to alternative methods?) appear to have inconsistencies in how many slabs were used. The CS method appears to, in practice, use 10-times more slabs than the other methods, making it unclear whether the improved performance of the CS method is due to it being a better method, or because the specimens were able to be measured with more precision (because of using more slabs).

Additional comments

See attached.

·

Basic reporting

The use of English is almost fine. I have revised some of the word usage and phrases. In many places the definite article ‘the’ is needed.

The introduction and background information are almost fine. The introduction does a good job of reviewing previous mass estimation methods, but fails on prior surface area estimations. This is unfortunate as the new method proposed in the paper does make and test surface area estimates. Two examples of the applications of animal surface area estimates to cite can be found in these two papers:

Henderson, D. M. (2013). "Sauropod necks: were they really for heat loss?" PLoS ONE 8(10): 1-8.
Sereno, P. C., et al. (2022). "Spinosaurus is not an aquatic dinosaur." eLife 2022: 1-15. (Figure 5)

The figures are almost fine. The dashed diagonals representing the diameters on figure 2 are confusing and took me a few seconds to realize what they were trying to show, and should be drawn differently. Also, the representative frustum and its subsection should be shown with same orientation as the body forms shown in figures 1 and 3.

Several sets of cross-sections from the full axial bodies and limbs from a limited selection of the models shown in figure 3 should be presented. The samples should include one each of a toothed and baleen cetacean, an ichthyosaur, a plesiosaur, the sturgeon and the turtle.

Experimental design

The research is appropriate for PeerJ. As the author states in the text, knowing the body masses for animals provides a foundation for other studies. The new cross-sectional method is described in sufficient detail to enable replication, and does produce valid results, and the testing and comparison of this new method appears to be adequate.

Validity of the findings

The results are reasonable. The conclusions are supported by the presented data.

Additional comments

At first glance the method does seem simpler and more direct than others, but the complexity of generating the initial 3D models in an expensive and complicated bit of software such as Rhino (used for the sources of the contours), and extracting the section contours is glossed over. It appears that the Rhino software used to produce the models already generates volumes and surface areas, so the new method seems a bit redundant.

The author has missed an opportunity to compute the centroid of the 3D shapes being analyzed. By adding an extra ‘l’ term in the integrand at the bottom of page 4, the centroid, , for the kth body segment could be determined with the following expression:

SEE ATTACHED PDF FOR THE EQUATION THAT DIDN'T TRANSFER TO THE WEB PAGE

This could be extended to determine the centroid for an entire body region. The centroid is useful to locating the balance point (if a density distribution is developed) or the centre of pressure for lift calculations (eg.for the underwater flying manta ray).

One serious limitation of the method is that it appears to depend on the body or body part being straight. Curved structures such as necks or tails would confound the simple, linear scaling scaling used. I would like to see if this new method could cope with the curved neck of a sauropod. See that attached image of a sauropod that could be used to test this. This image is from Henderson (2013) cited above.

The author presents his method, but it is up to the motivated reader to implement it. Most of the biologists and palaeontologists that I know who could use it could not even begin to develop the software to implement the method. Motani (2023), the author whose method is analyzed by the present author, does make his software available for use.

---

## Round 0.2 · Minor Revisions

The revised manuscript and additional examples make your analyses and comparisons easier to follow and of broader relevance. I would like to see this work published, but some minor but crucial points remain to be addressed:

1) You mention in the abstract and introduction that body mass and surface area is impossible to obtain in all extinct species. This is not entirely true – likely in most vertebrates but surface area can be calculated for extinct animal species with closed shells through CT (e.g., brachiopods, bivalves). Please rephrase.

2) Tyrannosaurus tail reconstruction: As highlighted by reviewer 2, the cross-sections of the tail completely neglect the presence of the substantial leg retractor muscle which leads to an underestimation of the tail mass and a more anteriorly located CM when compared to other studies (see suggestion provided by reviewer 2). Please use this suggestion or at least explicitly discuss this issue.

3) Add DOI to supporting materials on GitHub repository: As links on GitHub could change, please make sure an archive of your tutorial/supporting data on GitHub with digital object identifier is made and the DOI explicitly mentioned in the manuscript. This is crucial to be able to reproduce your study on the long-term. This can be done through Zenodo as far as I am aware: https://docs.github.com/en/repositories/archiving-a-github-repository/referencing-and-citing-content

4) Formatting: there are some additional formatting or typographical/language issues (compare suggestions of reviewer 2)

Please make sure to address these as well as all other points including those raised in annotated pdfs. I look forward to receiving the revised manuscript.

·

Basic reporting

I am satisfied that the revised manuscript has adequately addressed my concerns from the original submission. I am happy to support acceptance of this revision.

Experimental design

I am satisfied that the revised manuscript has adequately addressed my concerns from the original submission. I am happy to support acceptance of this revision.

Validity of the findings

I am satisfied that the revised manuscript has adequately addressed my concerns from the original submission. I am happy to support acceptance of this revision.

·

Basic reporting

This revised version of the original submission has responded to all of my previous comments, questions and suggestions.

The writing is clear and my requests for the addition of the definite article 'the' in many places in the text have been met.

The requested clarifications of the figures have also been done as requested.

The literature citations are all fine and bit more comprehensive than before.

Experimental design

The experimental design meets all the PeerJ requirements listed.

Validity of the findings

The results are consistent with the proposed methods and the 3D models provided as a test cases. All the PeerJ conditions for valid results and conclusions are met.

Additional comments

I only have one significant correction to make:

The cross-sections of the tail of Tyrannosaurus presented in Figure 6 completely neglect the presence of the substantial leg retractor muscle that lies along the proximo-ventral two-thirds of the tail - the caudifemoralis longus. This muscle is present in all limbed diapsids (eg. lizards, crocodilians, non-avian dinosaurs) Neglect of this muscle in the model results in an underestimation of the tail mass and a more anteriorly located CM when compared to other studies. See the attached JPEG image of what the base of the tail should look like with the proper musculature.

There are also a handful of wording corrections to make as well. I have indicated this in the annotated review PDF which is attached.

---

## Author Rebuttal · Round 0.2

# Response Letter to Editor and Reviewers

## Estimating body volumes and surface areas of animals from cross-sections

Dear Editor and Reviewers,

Thank you for your constructive comments. These comments are all valuable and helpful for improving my article. I have revised the article taking into account all the comments. Below I clarity the changes I made to the manuscript and respond to the questions raised by the reviewers.

## General Response

My apologies that I didn't show how the cross-sectional method (CSM) can be used in the original manuscript, and unfortunately this caused confusion. I have included in the revised manuscript what I want to convey in the following two paragraphs.

The validation was based solely on 3D models, and this may give the reader the impression that the CSM requires a 3D model to get the cross-sections. But for extant animals, the cross-sections can be acquired through dissection or CT scanning. This process might be time-consuming, but once the cross-sections of one individual are obtained, the CSM can be applied to study the body sizes of wild populations of this species. In such cases, the CSM would require only one photo rather than two required by other 2D approaches, and the results will be more accurate. I would agree that the CSM is more difficult to implement compared to other 2D methods, but this is a compensation for accuracy. While existing 3D models can be used to estimate the sizes of some species, for many animals we do not have 3D models of them. As shown in the article, assuming regular-shaped cross-sections may incorporate serious errors. Hence it is essential to assess the cross-sections when constructing 3D models. From this point of view, 3D modelling is more difficult to implement than the CSM since it can be time-consuming to adjust the details precisely.

For extinct animals, accurate skeletal reconstructions are essential to body mass or surface area estimation, as argued by some researchers like Gregory Paul (see the manuscript for citations). Previous studies have shown how to construct the cross-sections of the ribcages of extinct vertebrates (four publications are cited in the revised manuscript as examples). From this perspective, it is actually easier to meet the requirements of the CSM than those of other approaches. For example, GDI requires one side view and one dorsal/ventral view image of the skeletal reconstruction. Although the CSM requires a side view image and a series of cross-sections, it would be easier to construct multiple cross-sections than the entire dorsal view skeleton. In addition, it is generally easier to construct models and adjust the amount of soft tissue in 2D environments than in 3D software. Therefore, the CSM can be regarded as an alternative to 3D modeling under certain conditions while ensuring the reliability of the results.

To help the readers understand the applying scope of the CSM, a new section "work examples" was added to the revised manuscript (lines 186-231). This section contains an extant example

(bottlenose dolphin) and an extinct one (*T. rex*). The Discussion has been rewritten as well, with more details about the limitations of the CSM included.

# Editor

You provide a new way to estimate body volumes and surface areas from cross-sections and test its performance. This approach is of interest for the community but there are crucial points which need to be addressed before publication. I apologize for the delay in getting my decision to you, but I was hoping to receive an additional review which got an extension but unfortunately did not arrive. However, I feel that the input of the reviewers and my own should be sufficient to revise the manuscript.

Thank you for your evaluation of the manuscript and effort to invite reviewers.

## Background on competing approaches:

I agree with reviewer 1 that more information and comparisons on the competing approaches needs to be provided for the readers not so familiar with these approaches in the introduction and discussion. See also the recommendations of reviewer 2 concerning particularly prior surface area estimations.

Thank you for your comment. I agree that the introduction part should be revised. I have added more details about GDI and Paleomass in the introduction as suggested by Reviewer 1 (lines 55-65; lines 71-79). Some previous techniques to estimate surface area were also introduced as suggested by Reviewer 2 (lines 68-70; lines 86-87). More details about these changes can be found in the response to the reviewers below.

## Validity of findings:

Your test seem to produce valid results and comparisons seems adequate at first glance (compare reviewer 2) but the reasons for their performance are not fully explored. As pointed out by reviewer 1, the test of the methods lack generality and have inconsistencies. Please repeat test 1 by using more specimens, and using entire specimens as suggested by reviewer 1. Comparisons between methods have inconsistencies as more slabs are used in CS than other methods making it difficult to assess if it is truly a better methods or because specimens were able to be measured with more precision (compare reviewer 1).

I repeated test 1 on four entire models (lines 153-159), and the results are shown in Figure 7 of the revised manuscript. Test 1 aims to find out how many subslabs are required when partitioning a slab. In the revised manuscript, I added another test (labeled as test 2) to reveal how many slabs are required for each animal (lines 158-161). In test 3 and test 4 (the original test 2 and 3), all models were partitioned into 120 slices for GDI and CSM (120 slices for GDI, 12 slabs $\times$10 subslabs for the CSM).

For Paleomass, please see my response to Reviewer 1.

## Scientific reproducibility and accessibility:

The new cross-sectional method is described in sufficient details to enable replication pending considerable effort. You present a new method, but the software does not appear to be pub-

licly available and no clear guidelines or limitations for its use are provided (compare reviewer 2). This makes it hard to reproduce your approach and widely apply it for other taxa in a straightforward way (compare reviewer 2).

Currently it's beyond my ability to make this method fully automated in open source platforms (e.g., develop an R package for this), and the implementation of the CSM is very dependent on CAD software. Although the AutoCAD I used is a proprietary software, there are other alternative free CAD programs that can also do the job. You and Reviewer 2 offered a very good suggestion that this method can be improved by accomplishing automation in open source platforms in the future. To help users replicate the CSM in the current stage, I wrote a step-by-step tutorial (https://github.com/Pliosaurus-kevani/Cross-sectional-Method; This link is included in the manuscript: line 241).

Particularly critical is the complexity of generating initial 3D models in an expensive and advanced software not accessible to all (e.g., Rhino; compare reviewer 2). Reviewer 2 also points out that Rhino already generate volumes and surface areas, which makes your approach a bit redundant unless you can make the entire process available in a publicly available software (e.g., Motani 2023 - which method you analyzed - makes his software available for use).

Please see the general response section, the CSM doesn't necessarily need a 3D model to obtain the cross-sections.

As pointed out by reviewer 2 your approach could be easily extended by computing the centroid of the used 3D shapes or entire body region to locate the balance points or center of pressure for life calculations.

Yes, I have added the formulae about computing the center of mass (Eq. 11-13). The limitations of the CSM in computing the center of mass are stated in the Discussion (lines 357-359). For more details, please see my response to Reviewer 2.

The limitation of your approach that it depends on the body or body part being straight and how (if at all) it can be extended to cope with curved necks should be highlighted and discussed (compare reviewer 2).

I would agree that the CSM is not good at dealing with curved object, but there are some methods to solve this issue (lines 360-367). For more details, please see my response to Reviewer 2.

## Formatting and language issues:

Once revised, please check your manuscript for any inconsistencies (see suggestions by reviewers 1 and 2).

Thank you for being patient with my lack of proficiency in English. I have revised the manuscript following the suggestions of the reviewers. I hope the revised version meet the requirements for publication.

## Figures:

Please follow the recommendations of reviewer 2 to rework the dashed diagonals representing the diameters on Figure 2 and show the representative frustum and its subsection with same orientation as the body forms shown in figures 1 and 3. Please also follow the recommendation of reviewer 2 to present several sets of cross-sections from the full axial bodies and limbs from a limited selection of the models shown in figure 3 and to include in samples one each of a toothed and baleen cetacean, an ichthyosaur, a plesiosaur, the sturgeon and the turtle.

The slab and subslabs in Figure 2 were redrawn according to the suggestion of Reviewer 2. A figure containing several sets of cross-sections of the suggested samples was added to the revised manuscript as Figure 4.

# Reviewer 1 (Philip Novack-Gottshall)

## Basic reporting:

Generally well written but would benefit from editorial assistance with a fluent English writer to clean up numerous typos, grammatical errors, and occasionally non-standard sentences. Background information on the two competing methods (GDI and Paleomass) is insufficient at introducing them to readers, with critical information placed instead in the Conclusions.

Thank you for being patient with my lack of proficiency in English. I have revised the manuscript following the suggestions of yours. I also moved the description of the two competing methods in the latter part of the original manuscript to the Introduction section (lines 55-65; lines 71-79).

## Experimental design:

The overall design is fine. However, see below for concerns on the validity of the findings, which bear on the design.

Thank you for your evaluation. I will clarify the changes below.

## Validity of the findings:

Test 1 (how many slabs to cut the digital specimen into?) is lacking in generality; it should be repeated using more specimens, and using entire specimens. Tests 2 and 3 (how does CS compare to alternative methods?) appear to have inconsistencies in how many slabs were used. The CS method appears to, in practice, use 10-times more slabs than the other methods, making it unclear whether the improved performance of the CS method is due to it being a better method, or because the specimens were able to be measured with more precision (because of using more slabs).

Thank you for your comment, and this helps improve my experimental design. I have added more details on the purpose of each test. The Test 1 aims to find out how many subslabs to cut each slab into (lines 153-159). The CSM uses linearity to approximate non-linearity at small scales, and this test tries to find out how dense the partition needs to be. I added a new test (as Test 2) to clarify how many slabs to cut the specimen into (lines 160-163). I repeated Test 1 and Test 2 on the entire bodies of four animal models. It turns out that the number of slabs

for each animal and number of subslabs for each slab required to generate stabilized estimates vary across species, but 12 slabs, with each of which sliced into 10 subslabs, are sufficient for all the four samples tested. These two tests were conducted based on a limited sample size, hence I provided some additional recommendations for real-world applications in the Discussion (lines 314-325).

For Tests 3 and 4 (the original Tests 2 and 3), all models were partitioned into 120 slices for GDI and the CSM (120 slabs for GDI, and 12 slabs ×10 subslabs for the CSM). My apologies that I didn't describe Paleomass in great detail and caused confusion. Paleomass reads bitmaps, and creates a superellipse for each pixel along the sagittal axis (added in lines 73-79). It is technically impractical to accurately control how many pixels each silhouette takes up in a bitmap, but I set the resolution of pictures very high (each bitmap contains 6000 ×4000 pixels) following Motani (2023) (line 248). This means that each sample was actually sliced into several thousands of slabs in Paleomass. If the CSM provides more accurate results despite lower precision, then its benefits can be highlighted.

## Additional comments:

**Overview:** This manuscript presents a new method (cross-sectional method [CS]) for estimating the volume and surface area of animals, comparing it to two widely used methods (graphic double integration [GDI] and Paleomass). Three tests are implemented to demonstrate that the new method is both more accurate and precise than the other two. The strengths of the manuscript are a relatively simple (but powerful) method and its implementation to a wide number of marine vertebrates, spanning a well-chosen range of shapes (whales and ichthyosaur that can be well approximated by elliptical cross sections plus turtle and manta ray that are decidedly not). The weaknesses are primarily in its presentation and insufficiency of the tests. The manuscript should be revised to be more clearly organized (the background that introduces prior methods is too short). The first test is also too trivial to ensure the patterns are generally true, and this test should be repeated for additional samples. The other tests are appropriate, but it is unclear whether the number of slabs used across methods was correctly chosen, which makes it difficult to know if the results truly reflect methodological differences or instead how the methods were carried out.

Thank you for your comment. I have clarified some changes above, more details can be found below.

**Organization and writing:** The majority of the manuscript is well written. Much of the writing could be improved by using a native English speaker to correct the numerous typos and grammatical errors. A few examples are listed below.

Thank you for your help. I have corrected these errors.

Mostly importantly, more introduction is needed of how the CS method differs from GDI and Paleomass. Lines 55-64 are insufficient. Text in lines 215–235 is better, and most of that background should be moved to the Introduction.

Agreed. I moved that background to the Introduction and added more details (lines 55-65; lines 71-79).

The text seems to ignore the main weakness of the CS method, that it requires a complete

3D digital model, which is not always easily available. Because GDI and Paleomass assume the cross-sectional shape of most animals is elliptical (or hyperelliptical), they do not require perfectly complete 3D digital models. However, the CS method, because it measures cross-sectional area directly, appears to require a complete 3D digital model. Yes, the manuscript has convinced me that the new CS method appears to be an improvement over prior methods. However, if one has a complete digital model, why is ANY method needed? It can be directly measured in Rhino 7 (as was noted in line 163, and was used to obtain the "true volume" and "true area" in Supplementary Tables). I do not see how CS offers any improvement compared to measuring directly in 3D software. If there are benefits to CS over this alternative, they should be explained.

Please see the General Response section of this letter. In addition, I changed the terminologies "true volume" and "true area" to "observed volume" and "observed area".

**Tests:** Test 1 seems incomplete as a test of how many slabs are generally required for the CS method. Why only use a small portion of a sturgeon (and a portion that is quite morphologically similar throughout)? Using an entire animal seems more informative to provide a real-world recommendation. Consider adding additional models, and use the entire animal each time, so that a more realistic recommendation can be made for future users. Also, line 185 concludes that the errors stabilize at "10 or more" slabs, but figure 4 clearly shows stability (for this shape) at just 6 slabs. There should also be a sentence added somewhere to relate that this stability at 10 is the reason the author then used 10 slabs for subsequent tests.

I repeated Test 1 and (the new) Test 2 on four complete animal models. Details can be found above and in the revised manuscript. I also added why 12 slabs (each of which is further sliced into 10 subslabs) are used for Tests 3 and 4 in the Discussion (lines 345-347).

It seems there is variation in how the 3D models were cut into slabs. GDI and Paleomass used 10 slabs per specimen, whereas the CS method also used 10 slabs, but then cut each slab into 10 subslabs (making 100 total estimates). How much of the improvement in the CS method (compared to GDI and Paleomass) in tables 2 and 3 is caused by using an order of magnitude more slabs? To make a fair comparison, the number of slabs whose volume and area are estimated must be the same across methods. If subslabs are required for CS, then it seems there should be 1 slab in the first cut (i.e., the entire specimen) and 10 subslabs. Alternatively, there should be 100 slabs used for GDI and Paleomass.

Thank you for this comment, which helps improve the experimental design. I modified the numbers of slices for GDI and CSM, but didn't change the original data and results of Paleomass. The detailed changes and reasons have been stated above. The calculated results of GDI and the CSM changed slightly, while the final conclusion remains the same.

If you really want to highlight the benefits of your method, why not apply PaleoMass and GDI to the very non-elliptical shapes of the sturgeon, ray, and turtle? The third test only compares them to the fins of two of these (albeit the entire body for the sturgeon). I expected this result as soon as I saw the chosen shapes in figure 1. I think it would offer the best possible case to demonstrate the superiority of CS to ellipseassuming alternatives.

You are certainly right. I tested GDI, Paleomass, and the CSM on entire bodies of the sturgeon and the turtle. It turns out that the CSM has better performances than the two competing methods (Table 2 in the manuscript). The mouth of the manta ray model is open and connected

to a large cavity inside its body. GDI and Paleomass are not able to calculate the volume of this cavity because it can not be reflected in the side/dorsal/ventral view silhouette. In this case, it is difficult to tell whether the error originates from elliptical/superelliptical approximation or the presence of this cavity. This is why I tested only the fin rather than the whole manta model.

**Figure 1:** Consider reversing parts C and D horizontally, so that the anterior end is to the right side (to match the orientation in parts A and B). Unless I am mis-interpreting the shapes.

Thank you. I have made the changes according to your suggestion.

# Reviewer 2

## Basic reporting:

The use of English is almost fine. I have revised some of the word usage and phrases. In many places the definite article 'the' is needed.

Thank you for helping me find typos and grammatical errors sentence by sentence.

The introduction and background information are almost fine. The introduction does a good job of reviewing previous mass estimation methods, but fails on prior surface area estimations. This is unfortunate as the new method proposed in the paper does make and test surface area estimates. Two examples of the applications of animal surface area estimates to cite can be found in these two papers:

Henderson, D. M. (2013). "Sauropod necks: were they really for heat loss?" PLoS ONE 8(10): 1-8.
Sereno, P. C., et al. (2022). "Spinosaurus is not an aquatic dinosaur." eLife 2022: 1-15. (Figure 5)

Thank you for your comment to improve my manuscript. I have cited the two articles you mentioned in the Introduction (lines 68-70; lines 86-87). In addition, I described more details about how to estimate surface area using GDI and Paleomass. In the revised manuscript, prior techniques for surface area estimation are no longer contained in a separate paragraph that contains only one sentence, but described together with mass estimation methods.

The figures are almost fine. The dashed diagonals representing the diameters on figure 2 are confusing and took me a few seconds to realize what they were trying to show, and should be drawn differently. Also, the representative frustum and its subsection should be shown with same orientation as the body forms shown in figures 1 and 3.

Agreed. I have redrawn the Figure 2.

Several sets of cross-sections from the full axial bodies and limbs from a limited selection of the models shown in figure 3 should be presented. The samples should include one each of a toothed and baleen cetacean, an ichthyosaur, a plesiosaur, the sturgeon and the turtle.

Thank you for this suggestion. I added a new figure (Figure 4) to the revised manuscript showing some cross-sections.

## Experimental design:

The research is appropriate for PeerJ. As the author states in the text, knowing the body masses for animals provides a foundation for other studies. The new cross-sectional method is described in sufficient detail to enable replication, and does produce valid results, and the testing and comparison of this new method appears to be adequate.

Thank you for your positive comment.

## Validity of the findings:

The results are reasonable. The conclusions are supported by the presented data.

OK, thank you.

## Additional comments:

At first glance the method does seem simpler and more direct than others, but the complexity of generating the initial 3D models in an expensive and complicated bit of software such as Rhino (used for the sources of the contours), and extracting the section contours is glossed over. It appears that the Rhino software used to produce the models already generates volumes and surface areas, so the new method seems a bit redundant.

Please see the General Response of this letter and the work example section in the revised manuscript.

The author has missed an opportunity to compute the centroid of the 3D shapes being analyzed. By adding an extra 'l' term in the integrand at the bottom of page 4, the centroid, $\bar{l}$, for the kth body segment could be determined with the following expression:

$$\bar{l} = \frac{\int_0^{L_n} S_s l \, \mathrm{d}l}{\int_0^{L_n} S_s \, \mathrm{d}l}$$

This could be extended to determine the centroid for an entire body region. The centroid is useful to locating the balance point (if a density distribution is developed) or the centre of pressure for lift calculations (eg. for the underwater flying manta ray).

Thank you for your suggestion. I added three formulae about computing the center of mass (CM) to the manuscript (Eq. 11-13). To simplify the coumputation, I did not incorporate a coordinate system for the CSM. This unfortunately makes it impractical to compute the accurate position of the CM when only one set of cross-sections are available (i.e., only the vertical plane that contains the CM can be determined, and the height of CM can not be computed). I think this is a limitation of the CSM, which I mentioned in the revised Discussion (lines 357-359).

One serious limitation of the method is that it appears to depend on the body or body part being straight. Curved structures such as necks or tails would confound the simple, linear scaling scaling used. I would like to see if this new method could cope with the curved neck of a sauropod. See that attached image of a sauropod that could be used to test this. This image is from Henderson (2013) cited above.

Yes, the CSM is not good at dealing with curved objects because the calculation process requires the bases of each frustum being parallel. This is also acknowledged as a limitation in the

[Figure]

Discussion (lines 359-367). I also provided two methods to collect data so that the CSM can be applied to curved objects. It is recommended to acquire the vertical cross-sections of an animal. For extant animals, this issue can be avoided since in most cases the cross-sections obtained from CT scanning are vertically oriented. For extinct vertebrates, some previous studies have shown how to construct the vertical cross-sections of their ribcages (please see the manuscript for detailed citation).

For the sauropod you mentioned, the neck can be sliced as shown in this figure. If the vertical cross-sections can be reconstructed, then the CSM can be applied to calculate its volume and surface area. The neck of the *T. rex* example added in the revised manuscript is also curved, but I assume that the vertical cross-sections gradually change in shape from back of the skull to the ribcage. This can be viewed as part of the modeling process, since the true body outline of *T. rex* is not preserved.

Alternatively, the users can straighten the silhouettes prior to data collection. This figure is from a recent study by Motani and Pyenson (https://doi.org/10.7717/peerj.16978; the blue one is the original body reconstruction, and the black one is the modified). In general, I would

agree that the CSM is not good at handling curved objects. Some methods can be used to make it able to handle such objects, but this would take extra effort.

The author presents his method, but it is up to the motivated reader to implement it. Most of the biologists and palaeontologists that I know who could use it could not even begin to develop the software to implement the method. Motani (2023), the author whose method is analyzed by the present author, does make his software available for use.

Currently it is beyond my ability to make this method fully open source, but you offered a very good suggestion that this method can be improved by accomplishing automation in open source platforms in the future. To ensure that users can replicate the CSM in the current stage, I wrote a step-by-step tutorial of this method: https://github.com/Pliosaurus-kevani/Cross-sectional-Method. This link is included in the manuscript (line 241).

---

## Round 0.3 · accepted · Accept

Thank for addressing these final suggestions which include rephrasing/reformatting some aspects, adding the impact of musculature in the tail of Tyrannosaurus, and adding a DOI for the GitHub materials. I look forward to seeing this published.

·

Basic reporting

This is the review of the revised version. The author has addressed my single concern about the lack of musculature on the tail of the Tyrannosaurus that was originally supplied. The tail is now anatomically correct and the results coming from the analysis are consistent with previous findings.

Experimental design

The experimental design is all fine and meets PeerJ requirements.

Validity of the findings

The results appear valid given the methods and data used.

Additional comments

I have no further criticisms or suggestions to make with the manuscript.

---

## Author Rebuttal · Round 0.3

# Response Letter to Editor and Reviewers

## Estimating body volumes and surface areas of animals from cross-sections

Dear Editor and Reviewers,

Thank you for your comments. These comments are valuable and helpful for improving my article. I have revised the article taking into account all the comments. Below I clarify the changes I made to the manuscript.

# Eidtor

The revised manuscript and additional examples make your analyses and comparisons easier to follow and of broader relevance. I would like to see this work published, but some minor but crucial points remain to be addressed:

You mention in the abstract and introduction that body mass and surface area is impossible to obtain in all extinct species. This is not entirely true – likely in most vertebrates but surface area can be calculated for extinct animal species with closed shells through CT (e.g., brachiopods, bivalves). Please rephrase.

Thank you for your suggestion. I have revised the abstract and the introduction (lines 10 and 40).

## *Tyrannosaurus* tail reconstruction

As highlighted by reviewer 2, the cross-sections of the tail completely neglect the presence of the substantial leg retractor muscle which leads to an underestimation of the tail mass and a more anteriorly located CM when compared to other studies (see suggestion provided by reviewer 2). Please use this suggestion or at least explicitly discuss this issue.

Thank you for your comment. I have revised the *T. rex* model and reconstructed the tail cross-sections following a previous study. Two research articles on this topic are cited in the revised manuscript (lines 467, 489), and I have mentioned why the tail muscles should be taken into consideration (lines 216-218). The estimated CM of the revised model is consistent with the results published in previous studies (please see the article for citations; line 284). For more details, please see my response to Reviewer 2.

## Add DOI to supporting materials on GitHub repository

As links on GitHub could change, please make sure an archive of your tutorial/supporting data on GitHub with digital object identifier is made and the DOI explicitly mentioned in the manuscript. This is crucial to be able to reproduce your study on the long-term. This can be done through Zenodo as far as I am aware: https://docs.github.com/en/repositories/archiving-a-github-repository/referencing-and-citing-content

Thank you for your suggestion. I have created a DOI for the GitHub repository using Zenodo. The link is in line 243 of the revised manuscript.

## Formatting

There are some additional formatting or typographical/language issues (compare suggestions of reviewer 2)

Thank you for helping with the formatting. I have revised the manuscript taking into account of all your and Reviewer 2's comments. There is one thing that I need to explain, about "Benedon Paratodus" in the Acknowledgements section. Yes, this is a pseudonym. This person supported my work but wants to stay anonymous by using this name. I have explicitly stated this in the manuscirpt.

# Reviewer 1

## Basic reporting

I am satisfied that the revised manuscript has adequately addressed my concerns from the original submission. I am happy to support acceptance of this revision.

Thank you for your positive evaluation.

## Experimental design reporting

I am satisfied that the revised manuscript has adequately addressed my concerns from the original submission. I am happy to support acceptance of this revision.

Thank you.

## Validity of the findings

I am satisfied that the revised manuscript has adequately addressed my concerns from the original submission. I am happy to support acceptance of this revision.

Thank you for your positive comments.

# Reviewer 2

## Basic reporting

This revised version of the original submission has responded to all of my previous comments, questions and suggestions.

The writing is clear and my requests for the addition of the definite article 'the' in many places in the text have been met.

The requested clarifications of the figures have also been done as requested.

The literature citations are all fine and bit more comprehensive than before.

Thank you for your evaluation and comments.

## Experimental design

The experimental design meets all the PeerJ requirements listed.

Thank you.

## Validity of the findings

The results are consistent with the proposed methods and the 3D models provided as a test cases. All the PeerJ conditions for valid results and conclusions are met.

Thanks for the positive comments.

## Additional comments

I only have one significant correction to make:

The cross-sections of the tail of *Tyrannosaurus* presented in Figure 6 completely neglect the presence of the substantial leg retractor muscle that lies along the proximo-ventral two-thirds of the tail - the caudifemoralis longus. This muscle is present in all limbed diapsids (eg. lizards, crocodilians, non-avian dinosaurs) Neglect of this muscle in the model results in an underestimation of the tail mass and a more anteriorly located CM when compared to other studies. See the attached JPEG image of what the base of the tail should look like with the proper musculature.

Thank you for your comment and JPEG file, which improve my knowledge on dinosaurs. I found two research articles relevant to this theme and cited them in the revised manuscript: Persons and Currie (2011), and Snively et al (2019). Snively et al (2019) reconstructed the tail cross-section of FMNH PR 2081 according to an adult alligator. I investigated the caudal vertebrae of AMNH 5027 from the 3D model again. It turned out that the transverse processes of its caudal vertebrae are proportionally shorter than those of FMNH PR 2081. This seems to suggest a slimmer tail in AMNH 5027. Thus I reconstructed tail muscles following the criteria proposed by Persons and Currie (2011). Also, their criteria fit better with the JPEG file you sent me. The plane containing the CM of the revised model is located at the anterior edge of the pelvic girdle. This result is consistent with those from previous studies. Albeit the slimmer tail, the position of the estimated CM of AMNH 5027 is similar to that of FMNH PR 2081 reconstructed by Snively et al (2019). This is possibly because the ribcage of 5027 is thinner than that of 2081.

About the terminology: I checked the literature. It seems that both "m. caudifemoralis" and "m. caudofemoralis" are valid spellings. I used the latter following Persons and Currie (2011) and Snively et al (2019).

There are also a handful of wording corrections to make as well. I have indicated this in the annotated review PDF which is attached.

Thank you for your suggestions. I have revised the manuscript.